# Total Mercury and Fatty Acids in Selected Fish Species on the Polish Market: A Risk to Human Health

**DOI:** 10.3390/ijerph191610092

**Published:** 2022-08-15

**Authors:** Joanna Łuczyńska, Marek Jan Łuczyński, Joanna Nowosad, Monika Kowalska-Góralska, Magdalena Senze

**Affiliations:** 1Department of Commodity and Food Analysis, University of Warmia and Mazury in Olsztyn, Ul. Plac Cieszyński 1, 10-726 Olsztyn, Poland; 2Department of Ichthyology, Hydrobiology and Ecology of Waters, The Stanisław Sakowicz Inland Fisheries Institute in Olsztyn, Ul. M. Oczapowskiego 10, 10-719 Olsztyn, Poland; 3Department of Veterinary Prevention and Feed Hygiene, Warmia and Mazury University, 10-719 Olsztyn, Poland; 4ChemProf, 11-041 Olsztyn, Poland; 5Department of Limnology and Fishery, Institute of Animal Breeding, Faculty of Biology and Animal Science, Wrocław University of Environmental and Life Sciences, 51-630 Wrocław, Poland

**Keywords:** total mercury, fatty acids, marine and freshwater fish, quality indices, gutted fish, whole fish

## Abstract

The muscles of lake trout (*Salvelinus namaycush* Walbaum, 1792), crucian carp (*Carassius carassius* Linnaeus, 1758), flounder (*Platichthys flesus* Linnaeus, 1758), gilthead seabream (*Sparus aurata* Linnaeus, 1758), mackerel (*Scomber scombrus* Linnaeus, 1758) and tench (*Tinca tinca* Linnaeus, 1758) were examined. The total mercury (THg) was processed using the Milestone DMA-80 and the fatty acids were analyzed using the 7890A Agilent Technologies chromatograph. The THg content in analyzed fish ranged from 0.024 (lake trout) to 0.092 mg/kg wet weight (gilthead seabream). The muscles of fish examined had lower amounts of SFAs, and n-3 and n-6 PUFAs than MUFAs. The ratio of n-3/n-6 was higher in muscles of mackerel than other fish (*p* < 0.05). Due to the fact that both the THQ and HI are below 1, the tested fish are safe for the consumer from a nutritional point of view. Similarly, fatty acid indices indicate the safe consumption of selected fish species, and the daily consumption of the recommended dose of EPA + DHA (250 mg/day) and the concentration of mercury in fish calculations showed a hazard quotient for the benefit–risk ratio HQ_EFA_ below 1, suggesting that the intake of EPA + DHA poses no evident risk to human health. The ratio was calculated for a person weighing 60 kg. Therefore, it is important to monitor the fish, not only bought in the store, but also caught in various aquatic environments.

## 1. Introduction

Fish is a nutrient-rich food that varies in taste and texture, is versatile and low in saturated fat and, because of the low level of calories, is the perfect healthy diet food [1]. Fish has positive properties, as fish is one of the best sources of protein and the consumption of fish provides polyunsaturated fatty acids (PUFAs), liposoluble vitamins and essential minerals for human health [2]. Fish is softer and flakier than either mammals or poultry and the meat is inherently tender, because fish contains less connective tissue than beef [3]. At the same time, eating fish is a possible source of heavy metal exposure [4,5]. As the density is greater than 5 g/cm^3^, mercury is classified as a heavy metal [6] and is one of the rarest elements on Earth, ranking 74th on the list of 90 [7]. Hazardous and long-lasting contaminants in the environment, especially the aquatic environment, have been extensively studied [8]. For example, these pollutants can come from the atmosphere, from domestic and industrial wastewater [9,10], from human activities such as rapid industrialization, urbanization and other anthropogenic sources [11] or from direct or indirect natural sources (volcanic eruptions and weathering of metal-bearing rocks) [12]. Heavy metals, including mercury, introduced into the environment as a consequence of anthropogenic activities usually cause genotoxic damage in aquatic organisms [13]. Fish, through predation, control other aquatic organisms, mediate nutrient flows and can act as ecosystem engineers. As previously mentioned, when faced with increasing anthropogenic pressures, we must focus on the functions of fish that are related to their main ecological roles in the aquatic environment to address these problems [14,15]. Therefore, to avoid the increase in undesirable toxic metals in food intended for human consumption, it is necessary to limit industrial emissions and/or to limit concentrations in the lower links of the food chain, i.e., in water and soil, by setting tolerable levels [16]. This also applies to fish that live in deep waters, because consuming sea fish, in which there are toxic amounts of heavy metals, can also have a harmful effect on the human body [17]. Mercury is absorbed by fish and passed down the food chain to other predatory species at the end of the chain, which affects not only aquatic ecosystems but also humans through bioaccumulation [18]. Among vertebrates, they are unique and have two ways of obtaining metals: directly, i.e., from water through the gills, and through trophic absorption, i.e., from the diet through the intestines [19]. Ali and Khan [20] also argue that these metals can enter the fish’s body directly from water and sediment through the gills/skin, and from food/prey through the digestive tract. Grieb et al. [21] found that in order to protect human health related to the consumption of fish as the main route of human exposure to mercury, fish tissue monitoring programs have been established worldwide. Ali et al. [22] also reported the importance of monitoring metal concentrations in fish meat. This has to do not only with ensuring compliance with food safety regulations, but also with the resulting consumer protections [8,17,23].

Min et al. [24] reported that mercury and essential fatty acids can co-occur in dietary sources of fish, and both were conveyed along the aquatic food chain, but their concentration varied in different links of the aquatic food chain.

Docosahexaenoic acid (DHA), which is a polyunsaturated fatty acid (PUFA), is the principal constituent of a variety of cells, especially of brain neurons and retinal cells, and plays an important role in fetal brain development, the development of motor skills, lipid metabolism, visual acuity in infants and cognitive support; along with eicosapentaenoic acid (EPA) it plays an important role in preventing atherosclerosis, rheumatoid arthritis, Alzheimer’s disease and dementia [25]. Other authors reported that essential fatty acids were positively correlated with a reduction in cardiovascular morbidity and mortality, hypertension and diabetes mellitus [26], lowering plasma triglyceride levels and blood pressure [27] and contributing to various membrane functions (fluidity, permeability) [28]. Khora [1] wrote about a similar effect of these acids on the human body. In summary, long-chain (LC) polyunsaturated fatty acids n-3 (PUFAs), in particular DHA and EPA, are nutrients that are involved in many metabolic and physiological processes; n-3 LCPUFAs have been extensively tested for their impact on human nutrition and health [28]. It is known that the human body cannot synthesize certain fatty acids; hence, the essential fatty acids must be consumed in the diet, and fish and other aquatic organisms are the main sources of polyunsaturated fatty acids (PUFAs). Therefore, humans obtain most of the eicosapentaenoic acid (EPA) and docosahexaenoic acid (DHA) by consuming fish, aquatic invertebrates and algae [29]. Next to fish, vegetable oils (rapeseed or canola), linseed or flaxseed, olive, peanut, soya and walnut oils, green leafy vegetables, fenugreek seeds, kidney beans and dry fruits are the main dietary sources of n-3 PUFAs. Vegetable oils (sunflower, safflower, sesame, palm olein and corn oils) are rich dietary sources of n-6 PUFAs [30]. Long-chain polyunsaturated fatty acid (LCPUFA) biosynthetic pathways have been extensively studied in farmed fish, both for determining specific dietary essential fatty acids that guarantee optimal growth and development in captivity, and for maintaining high levels of healthy n-3 LCPUFAs for consumers [31]. Mohanty et al. [25] noted that due to the high availability of many species of food fish, consumers have a wide choice owing to availability and affordability. According to Taşbozan and Gökçe [29], the increasing demand for fish and the stabilization of marine fish and freshwater landings have contributed to a widening gap between demand and supply for fish and fish products. This leads to a necessity to improve aquaculture production.

The aims of our studies are:To determine the differences between the mercury content in the muscles of various fish species, both marine and freshwater;To estimate, on the basis of THQ and HI indicators, whether these fish are safe from a nutritional point of view, and thus can be consumed by humans;To estimate the fat content, profile of fatty acids and, the most important, lipid quality indices in muscles of fish species from Polish markets.

## 2. Materials and Methods

### 2.1. Sample Collection and Preparation

A total of 36 of the most important commercial fish belonging to six species include:

Gutted fish:
Tench (*Tinca tinca* Linnaeus, 1758) (*n* = 6);Lake trout (*Salvelinus namaycush* Walbaum, 1792) (*n* = 6);Flounder (*Platichthys flesus* Linnaeus, 1758) (*n* = 6).


Whole fish:
Gilthead seabream (*Sparus aurata* Linnaeus, 1758) (*n* = 6);Crucian carp (*Carassius carassius* Linnaeus, 1758) (*n* = 6);Mackerel (*Scomber scombrus* Linnaeus, 1758) (*n* = 6).

All fish were purchased from Olsztyn markets (northeastern Poland). Among the studied fish, there were also those from Baltic catches, i.e., the Polish catch area (flounder); harvested by fishermen from our lakes (tench and crucian carp) or lakes in Canada (lake trout); fish from ocean catches (mackerel—Atlantic Ocean) and sea catches (gilthead seabream—Mediterranean Sea). Before dissecting the whole fish, they were weighed (±1 g) and measured (±0.1 g). The muscles of whole and gutted fish were filleted from the dorsal part with a plastic knife and fork and then thoroughly mixed. The samples were performed on plastic, disposable plates and placed in polypropylene bags. Specimens of each species were stored at −30 °C until analysis. Samples were prepared in two parallel replications.

### 2.2. Chemical Analyses of Mercury

The samples were weighed in two parallel repetitions into quartz boats (approximately 100 mg ± 0.0001 g) and transferred to an analytical autosampler. The total mercury content was processed using the Milestone DMA-80 (with dual-cell) direct thermal decomposition mercury analyzer (Italy). The content of mercury was measured in peak area mode against a linear calibration curve (regression coefficient: R^2^ ≥ 0.999). The method parameters were described in the previous publication [32]. The detection limit (LOD) was 0.02 μg/kg, whereas the quality control of the method was assured using the reference material: BCR CRM 422 (muscles of cod *Gadus morhua* L.) with a certified mercury value. The recovery rate of Hg was 100.2% (*n* = 4).

### 2.3. Target Hazard Quotient (THQ)

The content of mercury was used for the estimation of the ratio between the exposure and the oral reference dose [33]. According to the USEPA [34], the oral reference dose (RfD) was 3.00 × 10^−4^ mg/kg/day.
THQ = (EFr × ED × FiR × C/RfD × BW × TA) × 10^−3^(1)
where EFr is the exposure frequency (365 days/year); ED is the exposure duration (70 years); FiR is the fish ingestion rate (g/person/day); C is the average concentration of heavy metals in food stuffs (μg/g wet weight); RfD is the oral reference dose (mg/kg/day); BW is the average body weight of local residents (60 kg) [35]; TA is the average exposure time (365 days/year × ED).

HI was calculated as the sum of THQ values:HI = THQ (foodstuff 1) + THQ (foodstuff 2) + THQ (foodstuff 3) + THQ (foodstuff 4) + ……(2)

### 2.4. Lipid Content and Fatty Acid Analysis

#### 2.4.1. Fat Content

Total lipids (samples of approximately 1 g ± 0.0001 g) were extracted with the help of the E-816 HE automatic extractor (BUCHI, Flawil, Switzerland) using organic solvent petroleum ether. Prior to analysis, duplicate samples were dried to a constant weight at 105 °C on filter papers, then transferred to weighed beakers equipped with glass thimbles. The hot extractions consisted of 3 steps: extraction, rinsing and drying. After the extraction, all of the solvent was collected in the tank. The total lipids were dried in a beaker at 100 °C to a constant weight and then weighed.

#### 2.4.2. Fatty Acid Analysis

The lipids were extracted according to the modified Folch method. The method was performed according to Christie [36]. The fatty acid methyl esters were prepared from total lipids with the Peisker method with chloroform: methanol: sulfuric acid (100:100:1 *v*/*v*) [37].

The fatty acids of methyl esters of each sample were carried out on the 7890A Agilent Technologies chromatograph (Agilent Technologies, Inc., Santa Clara, CA, USA) equipped with a flame ionization detector (FID) and capillary column (dimension of 30 m with a 0.32 mm internal diameter, liquid-phase StabilwaxR). The temperature ionization detector was set at 250 °C, the injector at 230 °C and the column at 190 °C. The carrier gas was helium with a flow rate of 1.5 mL/min. Individual fatty acids were identified by comparing the relative retention time peaks to the known Supelco standards.

The fatty acids were quantified in g/100 g of edible fish muscles according to the following formula [38,39]:FA (g/100 g) = [(P × FC)/100] × C(3)
where FA is the fatty acid (g/100 g muscles of fish); P is the fatty acid (% of total lipids); FC is the fat content (g/100 g of fish muscles); C is the conversion factor (0.900 for fatty fish).

### 2.5. Lipid Quality Indices

Calculations of the hazard quotient for the benefit–risk ratio were conducted according to Gladyshev et al. [40].
FP = R_EFA_/C(4)
DM = FP × c(5)
HQ = DM/RfD × AW(6)
HQ_EFA_ = (R_EFA_/C) × c × (1/(RfD × AW)) = (R_EFA_ × c)/(C × RfD × AW)(7)
where EFA is the EPA + DHA (250 mg per day for a human person is presented here as R_EFA_); C is the content of EFA (mg/g); FP is the fish portion a person consumes that will obtain a dose of metal; DM (µg per day); c is the content of metal in fish (µg/g); RfD is the reference dose of metal (3.00 × 10^−4^ mg/kg/day); HQ is the hazard quotient; AW is an average adult weight (60 kg); HQ_EFA_ is the hazard quotient for fish consumption when a person aims to obtain from the fish the recommended dose of EFA (risk benefit ratio for fish consumption of both metal and EFA).

The lipid quality indices (AI and TI) were calculated using the following formulas by Ulbricht and Southgate [41], Garaffo et al. [42] and Telahigue et al. [43]:

Index of atherogenicity (AI):AI = [C12:0 + (4 × C14:0) + C16:0]/(n-3PUFA + n-6PUFA + MUFA)(8)
Index of thrombogenicity (TI): TI = [C14:0 + C16:0 + C18:0]/[(0.5 × C18:1) + (0.5 × sum of other MUFA) + (0.5 × n-6PUFA) + (3 × n-3PUFA) + n-3PUFA/n-6PUFA)](9)
FLQ = 100 × [EPA + DHA]/[% of total fatty acids](10)
Hypercholesterolemic fatty acids (OFAs): OFA = C12:0 + C14:0 + C16:0(11)
Hypocholesterolemic fatty acids (DFAs): DFA = C18:0 + UFA(12)

### 2.6. Statistical Analysis

Analysis of the results was performed using Microsoft Office Excel 2019 and Statistica 13.3. (StatSoft, Inc., 2300 East 14th Street, Tulsa, OK, USA). All statistically significant differences were calculated at *p* < 0.05. Due to the data being defined as having a non-normal distribution, the Kruskal–Wallis test with post hoc analysis was used. An attempt was made to determine the value by allowing the data to be divided into two groups differing in a statistically significant manner. The results are presented when such a value could be determined. 

The PCA test using R-statistics was applied in order to visualize the differences between the groups (RStudio Version 1.1.442—© 2009–2018, RStudio, Inc., Boston, MA, USA). It was performed on the basis of all data: differences in the parameters of the examined parameters depending on fish species.

## 3. Results

### 3.1. The Differences between the Mercury Content in the Muscles of Fish Examined

The highest content of mercury was found in muscles of gilthead seabream, although these differences were statistically significant only between crucian carp and lake trout (*p* < 0.05) (Table 1). The muscles of lake trout **contained** the lowest **content of** mercury, but significant differences were found between mercury values in the muscle tissue of **lake trout and three other species** (gilthead seabream, mackerel and tench).

### 3.2. 2. THQ (Target Hazard Quotient) and HI (Hazard Index) in Muscles of Fish Examined

The THQ and HI of individual foodstuff were below 1 and ranged between 0.0473 (lake trout) and 0.1782 (Seabream), whereas the HI for a specific receptor/pathway combination exceeded 1 and was 0.684. 

### 3.3. The Differences between Profile of Fatty Acids (%) and the Lipid Quality Indices in Muscles of Fish Species

The saturated fatty acids (SFAs) varied between 13.78% (lake trout) and 32.17% (tench). The muscles of lake trout had statistically significant lower SFAs than other fish species (*p* < 0.05) (Figure 1). Palmitic acid (C16:0), which is a long-chain SFA, was the predominant fatty acid in this group as it had sixteen carbon atoms (Table 2).

The opposite regularity was found in the case of monounsaturated fatty acids (MUFAs), because the muscles of lake trout (55.58%) contained more of these acids than the muscle tissue of tench (26.62%) and other fish examined (36.89–43.90%) (*p* < 0.05), whereas in the muscles of tench was found statistically significantly lower amounts than in the other fish studied (*p* < 0.05) (Figure 1 and Table 2). The muscles of fish examined had lower amounts of SFAs than MUFAs. Oleic acid (C18:1cis9) was the dominant monounsaturated fatty acid and ranged from 13.93% (tench) to 43.23% (lake trout) (Table 2). The muscle tissue of marine fish (flounder and mackerel; 27.06 and 25.31%, respectively) and freshwater fish (tench and crucian carp; 24.09 and 21.03%, respectively) had the higher values of n-3 PUFAs than gilthead seabream and lake trout (*p* < 0.05) (Figure 1). The muscles of gilthead seabream, which is a marine fish, contained significantly similar amounts of n-6 PUFAs as lake trout and tench, which represent freshwater fish (*p* > 0.05). The lowest value of n-6 PUFAs was observed in the muscles of mackerel (3.17%) compared to other fish (*p* < 0.05), whereas the muscle tissue of flounder had a similar content of these acids (7.24%) to the muscles of crucian carp (7.71%; *p* > 0.05) (Figure 1). Arachidonic acid (C20:4 n-6) and linoleic acid (C18:2 n-6) were the predominant polyunsaturated n-6 fatty acids, whereas among the n-3 fatty acids, docosahexaenoic (DHA) and eicosapentaenoic (EPA) fatty acids dominated, except in the muscles of lake trout and crucian carp (Table 2). The n-3/n-6 ratio in the muscles of mackerel was the highest (8.10) when compared with other fish studied. The n-3/n-6 ratio can be ordered in descending order: mackerel > flounder ≈ crucian carp > tench ≈ lake trout ≈ gilthead seabream.

The atherogenic index (AI), thrombogenicity index (TI) and flesh-lipid quality index (FLQ) were calculated as follows: 0.18–0.50 (AI), 0.16–0.34 (TI) and 7.07–22.76 (FLQ). Hypercholesterolemic (OFA) and hypocholesterolemic (DFA) fatty acids in muscles of fish ranged from 10.50 to 24.44 and from 73.93 to 88.61, respectively. In the case of these indices, no clear differences between fish species were found (Table 2).

The sum of EPA and DHA varied between 7.07% (lake trout) and 22.76% (flounder) (Table 2), and in between 0.762 (crucian carp) and 15.833 (mackerel) mg/g (Figure 2), is The hazard quotient for fish consumption (HQ_EFA_) ranged between 0.07 (mackerel) and 0.90 (tench) (Figure 3).

The PCA reported that there were significant differences in the content of fatty acids and Hg between the studied species of fish (Figure 4).

## 4. Discussion

### 4.1. Mercury Content in the Muscles of Various Fish Species

Of the 19 trace metals, mercury is classified as one of the potentially toxic elements [44,45]. The content of mercury is influenced primarily by species, but also by factors including age, environmental conditions, exposure time and concentration [46]. Jakimska et al. [47] reported that the bioaccumulation of heavy metals is due to age, body size and is mass dependent. In our publication we focused on two factors: species and body size (body length and weight). According to Saulité and Svecevičus [19], heavy metal accumulation depends on abiotic and biotic factors, which is confirmed by the research of other authors. The muscles of tench from the Vistula River (Poland) contained more mercury (0.34 mg/kg wet weight) [48] than crucian carp from this same region (0.14 mg/kg wet weight). Misztal-Szkudlińska et al. [49] found that the mercury content in the fish decreased in the following order: crucian carp > tench. Muscle tissue of crucian carp, belonging to the group of benthophages, from Warta Mouth National Park (west Poland) contained (0.099 mg/kg mercury) three times higher mercury content than the fish we examined (Table 1). Similar observations were made by Jovanović et al. [50] studying Prussian carp (*Carassius auratus gibelio*) from the Danube River (territory of Zemun, Belgrade, Serbia). In the case of fish belonging to a higher level of the food web, such as flounder (coastal zone of the Baltic Sea, Poland), mercury accumulation in the muscle tissue prevailed, which is related to biomagnification in the marine trophic chain [51]. Pokorska et al. [52] observed that differences in bioaccumulation between fish species are related to their affinity for metal uptake from the aquatic environment, as well as their place in the trophic chain. The content of mercury in muscles of fish from the Baltic Sea (Poland) decreased as follows: crucian carp (0.00655 mg/kg) > flounder (0.00393 mg/kg) (*p* < 0.05) [53]. In our studies, interspecific differences between the content of mercury in the muscle tissue of the same fish were not found (Table 1). According to Kalisińska et al. [54], mercury content was significantly different between predators and benthivorous fish. The muscles of crucian carp, belonging to the benthophage group, contained three times more mercury (0.099 mg/kg wet weight) than the crucian carp examined (Table 1). The frozen muscle tissue of seabream produced and marketed in Turkey [55] had lower values of mercury (0.0749 mg/kg) than the seabream we studied (Table 1). Our research found that the mercury content in lake trout muscles (Table 1) was similar to those in lake trout muscles collected across Canada [56]. The same authors also found that mercury content increased significantly with the trophic level, and the lake trout is a representative of predatory fish, occupying the last place in the trophic chain of the aquatic environment. The content of mercury in muscles of Atlantic mackerel, which was purchased in the Jagalchi fish market of Korea [57], also was similar to that recorded in the fish covered by the studies (Table 1). The interspecific differences in mercury level in the muscles of flounder and mackerel examined (Table 1) were not statistically significant (*p* > 0.05). These results are consistent with previous studies [58]. Mackerel, as a representative of pelagic fish from the southeastern and northwestern Sicilian coast, had a lower amount of Hg (0.298 mg/kg) than seabream (demersal fish) (0.455 mg/kg) [59].

### 4.2. Human Health Risk Assessment

Naccari et al. [59] found that the mercury concentration in the fish from the northwest coast of Sicily was higher than the PTWI, which may pose a threat for the local population; therefore, the consumption of this fish needs to be monitored. Next to these fish, various living organisms, such as microbes, fungi, plants, animals and humans, are used to monitor toxic metals from the air, water, sediments, soil and food chain [60].The target hazard quotients in the muscles of flounder, cod and crucian carp from the Baltic Sea (Poland) were also below one; hence, the authors concluded that the consumption of these fish poses no risk [53]. This is in accordance with our studies (Table 1). The research by Kareem et al. [61] revealed that the heavy metals investigated, including mercury, in the organs and flesh of flounder (except liver) was above the limits recommended by the World Health Organization. The THQ for the muscles of fish from the Vistula River (Poland) was also below one (0.4) [48]. This is in accordance with our studies (Table 1). When the THQ < 1, there is a health benefit from fish consumption and the consumers are safe, whereas a THQ > 1 suggests a high probability of adverse risk to human health. Due to the fact that the mercury content in seabream fillets (Turkey) did not exceed the limit according to EC 188/2006, the authors concluded that these fish were of good quality [55]. For edible tissues of mackerel from the Azores archipelago, the THQ was also <1 [5]. These results are consistent with our studies (Table 1). According to above authors, the total mercury present in carnivorous species is significantly higher than the Hg observed in omnivores at all sampling sites. No significant differences were found between FAO fishing zones. The same authors reported that fish species feeding on organisms at higher trophic levels show a significantly higher total Hg than fish species feeding at lower trophic levels, suggesting biomagnification trends. It could be observed that demersal species contain higher contents of Hg than pelagic species. It is not possible to draw such unequivocal conclusions in our research. The fish came from different reservoirs, and it is known that the mercury content in fish is influenced by a number of factors, not only the species. In the case of mackerel from different parts of Peninsular Malaysia, the estimated weekly intake of total mercury from the consumption of 66.33 g per week was lower than the maximum limit established by the FAO/WHO and codex [62]. The concentration of mercury in the tissues of mackerel from a frozen fish market (Nigeria) was above the recommended safety limits outlined by the FAO/WHO [63]. These results are not in accordance with our studies (Table 1). Performed by Amariei et al. [64], the evaluation of fish, including cod, showed that marine organisms are safe for the consumer in terms of mercury content. The authors compared the mercury content with the values included in Commission Regulation (EC) No 1881/2006 and the PTWI established by experts of the WHO/FAO Joint Committee. Rodríguez et al. [65] found that by consuming 350 g of pangasius per week, the contribution to the tolerable weekly intake (TWI) of mercury was 32% and 27.5% for women and men, respectively. At the same time, the authors stated that there should be a restriction on children’s consumption. In the case of pangasius, hake and seabream from the central market of Granada (Southern Spain), the content of mercury in muscles did not exceed the EC regulation [66]. In another case, it turned out that in Nile tilapia from Pra (Ghana), mercury exceeded the set limits of the WHO for human consumption for both children and adults, and hence, was not safe for consumption [67]. Based on the above literature data, it can be concluded that, depending on the place of harvesting which could be related to the pollution of a given aquatic environment or accumulation of mercury in the food chain, the quality indicators and limits for mercury content in fish from a given reservoir could be exceeded. Some data showed that there were species of fish whose consumption poses no risk to human health.

### 4.3. Fatty Acid Composition

Aquatic organisms (fish, invertebrates, algae and other aquatic foods) are the main natural sources of polyunsaturated fatty acids in the human diet, but fish are the main contributors of n-3 PUFAs in human nutrition [29,68]. According to Taşbozan and Gökçe [29], the fatty acid composition of fish differs depending on species (interspecific and intraspecific), feeding habits, their aquatic environment (marine or freshwater, cold or warm water, salinity, temperature), seasonal changes, size, migration, sexual maturity and environmental conditions. In the muscles of mackerel and flounder, as representatives of marine fish, higher values of n-3 PUFAs were found than in the muscle tissue of other fish studied (Figure 1), whereas the muscles of gilthead seabream, belonging to the same group but living in a different body of water, characterized the lower content of n-3 PUFAs. The flatfish, including flounder, are the sea fish most popular with consumers. Unlike sea fish, the catch of freshwater fish that includes tench is of little economic importance, although it does offer an important source of biologically valuable proteins [69].

Dellinger et al. [70] studied six fish species, including lake trout, and noted intra- and interspecies variability. In another example, a strong relationship between feeding behavior and fatty acid composition of the muscle lipids of fish from subalpine lakes was observed by Vasconi et al. [71]. The same authors found that the amounts of n-3 PUFAs varied depending on the fish species, the lipid content and the feeding habits, as well as varying in different lakes and months of capture. Planktivorous fish, which belong to the pelagic habitat, had the lowest n-3 and n-6 PUFAs, but contained more MUFAs. In omnivorous fish, including tench and crucian carp, the following items were found in the digestive system: vegetal and animal food, a substantial proportion of n-3 PUFAs and the highest value of n-6 LCPUFAs and n-6 fatty acids. All fish examined, with the exception of tench (Figure 1), had more MUFAs, whereas the muscles of tench contained higher amount of SFAs. These results were not consistent with the observation noted by Vasconi et al. [71]. Polat et al. [72] wrote about the influence of the season on the content of fatty acids in tench captured from Seyhan Dam Lake (Turkey). These authors reported that the sum of PUFAs in November (41.85%) is higher than in July and August, which has to be considered in diet preparation. These values are similar with those in the current study for the same species (Table 2), whereas the sum of EPA and DHA in the muscle lipids of tench from the present study ranged between the values of 9.29–22.47% observed by the above authors. Similarly, Branciari et al. [73] identified and quantified a total of 23 major fatty acids, including eight SFAs, six MUFAs and nine PUFAs. The same authors reported that the all fatty acids varied significantly among species. There was also tench among the fish studied, which is regarded as a low-fat fish (2–4% fat). In conclusion, the authors found that muscles of species such as eel (*Anguilla anguilla* Linnaeus), sand smelt (*Atherina boyeri* Risso), carp (*Cyprinus carpio* Linnaeus) and tench had a higher content of n-3 long-chain EPA and DHA.

Other authors [74] observed that the fatty acid profile of seabream reflected the fatty acid contents of the feeds used in their feeding. The muscles of wild gilthead seabream contained more n-3 PUFAs (38.79%) and a higher n-3/n-6 ratio (3.91) than the farmed seabream (22.71 and 26.12%, 1.24 and 2.04, respectively) [75]. The authors observed a reverse regularity for n-6 PUFAs (10.95% for wild fish and 13.91 or 19.23% for farmed fish). These results for n-3 PUFAs and the n-3/n-6 ratio were higher than those of the same fish species in the current study (Table 2, Figure 1). Lenas et al. [76] compared the fatty acid composition of wild and cultured gilthead seabream from a lagoon in NW Greece and found that muscles of farmed fish contained higher levels of MUFAs, n-3 PUFAs and n-6 PUFAs (39.47%, 19.89% and 12.20%, respectively) than wild fish (37.67%, 15.87% and 7.21%, respectively). The ratio of n-3/n-6 and SFAs were lower in farmed fish. Considering the content of total PUFAs, Amoussou et al. [77] noted that in contrast to SFAs and MUFAs, cultivated seabream contained higher PUFAs than wild fish. Therefore, seabream next to seabass (*Dicentrarchus labrax* Linnaeus) cultivated in Corsica (Mediterranean Sea) are seafood with good nutritional qualities for human health. Linhartova et al. [78] investigated important freshwater fish species from semi-intensive and extensive culture systems (including tench) and noted that the muscles of all fish are of high nutritional quality. The same authors demonstrated that the AI and TI ranged from 0.27 to 0.63 and 0.20 to 0.61, which confirmed a great benefit for human health and provided important nutritional information about these fish species. The muscles of tench studied by these authors had a 0.42 AI (semi-intensive and extensive systems) and a 0.33 or 0.27 TI (semi-intensive and extensive systems, respectively). These values are similar to those obtained in this study for the same species (Table 2). The sum of EPA and DHA varied between 0.762 (crucian carp) and 15.833 mg/g fresh meat (mackerel) (Figure 2). The ratio of benefits of consuming the recommended dose of DHA + EPA to the risk of mercury consumption shows that an HQ_EFA_ < 1 indicates the health benefits of fish consumption, and an HQ_EFA_ > 1 means risk (Figure 3).

The differences in the chemical composition of fish, including fatty acids, are related to the fish species, but are also conditioned by other individual and environmental factors. The composition of lipids also depends on the type of fish tissue they come from (muscles, roe, organs). The main lipid-storing organs in fish are the muscles and the liver, although the liver alongside the roe (caviar) or other organs are seen in different regions and countries as examples of traditional foods of local importance [79]. 

Twenty-three fatty acids in the muscle of seven freshwater fish species (China) were identified and compared by Hong et al. [80]. Saturated fatty acids (SFAs) in the white muscle of three omnivorous species (crucian carp; bighead carp, *Aristichthys nobilis* Richardson; and silver carp, *Hypophthalmichthys molitrix* Valenciennes) were significantly higher (*p* < 0.05) than carnivorous, planktivorous and other omnivorous fish. The main food sources for crucian carp are plankton, benthic invertebrate and plant materials. The above authors found that the C16:0 dominated the saturated fatty acids in the muscles of seven freshwater fish. Additionally, in the case of the studied fish species, C16:0 dominated the SFAs (Table 2). The typical sources of palmitic acid are most fats and oils [81]. This acid, along with other acids such as C12:0 (lauric) and C14:0 (myristic), belongs to the hypercholesterolemic fatty acids (OFAs) and increases LDL cholesterol, whereas stearic acid (C18:0), belonging to the hypocholesterolemic fatty acids (DFAs), has no effect [81]. Typical sources of C12:0 fatty acids are coconut, oil and palm kernel oil, whereas C14:0 is also a dairy fat. If we replace SFAs (C12:0-C16:0) with PUFAs, then LDL cholesterol and the total/HDL cholesterol ratio will decrease. Hence, it is known that replacing SFAs with PUFAs decreases the risk of coronary heart disease (CHD) and there is convincing evidence for this, whereas replacing SFAs with MUFAs reduces the LDL cholesterol total/HDL cholesterol ratio [81]. The muscles of crucian carp (China) contained the groups of fatty acids as follows: SFAs (32.53%) > n-6 PUFAs (27.86%) > MUFAs (26.58%) > n-3 PUFAs (10.76%). The muscle tissue of the same species in the present study (Table 2) characterized the content of n-3 PUFAs as twice as high of that observed by Hong et al. [82]. Due to the recommendations of the FAO [81], it is suggested to change the consumption of saturated fats to unsaturated fats. It was also recommended to limit the level of saturated fat in the diet [82].

## 5. Conclusions

The fish studied belong to different links in the trophic chain, but at the same time they were purchased in two forms. One of the factors determining the mercury content of the fish is the type of feeding, depending on the species. In the study, there were few differences between the mercury content in the muscles of the studied fish species. This could be due to the fact that these fish came from different water bodies, and the mercury content is influenced by a number of abiotic and biotic factors. 

Looking at the low levels of mercury in the muscles of the studied species of fish, as well as the THQ and HI < 1, it can certainly be said that consumption of meat from all the species examined is safe from a health point of view, and daily consumption of the recommended dose of EPA + DHA (250 mg/day) and the concentration of mercury in fish calculations showed a hazard quotient for the benefit–risk ratio HQ_EFA_ below 1, suggesting that the intake of EPA + DHA poses no evident risk to human health. The ratio was calculated for a person weighing 60 kg.

In all fish species, being the aim of this current study, the atherogenic index (AI) and thrombogenicity index (TI) were below one, whereas undesirable hypercholesterolemic fatty acids (OFAs) were less than the hypocholesterolemic fatty acids (DFAs) that have a desirable dietary effect, which confirms that eating the fish examined is not detrimental to human health.

Based on the above statements, the consumption of all these species is safe for the consumer from a nutritional point of view. The best source of EPA and DHA was flounder and mackerel, which are popular on the Polish market.

Therefore, it is important to monitor the fish, not only those finally bought in the store, but also those caught in various aquatic environments where our fish come from, i.e., Baltic catches, the Polish catch area (flounder); harvested by fishermen from our lakes (tench and crucian carp) or lakes in Canada (lake trout); fish from ocean catches (mackerel—Atlantic Ocean) and sea catches (gilthead seabream—Mediterranean Sea).

## Figures and Tables

**Figure 1 ijerph-19-10092-f001:**
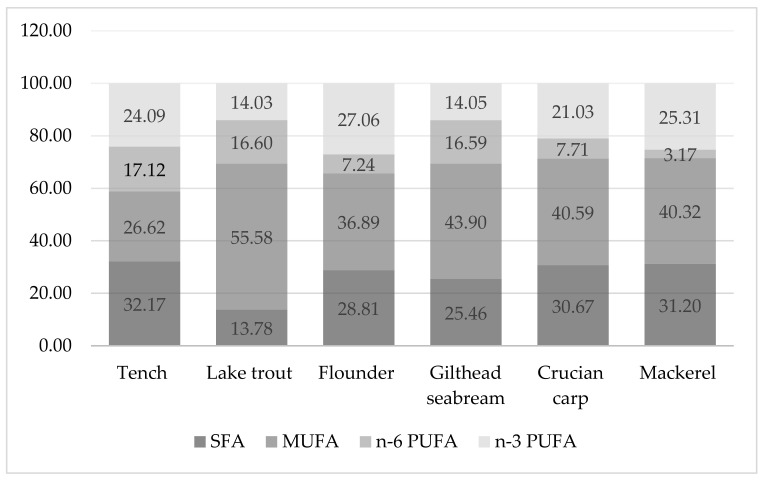
The percentage of fatty acid groups in the total fat composition of fish examined (%).

**Figure 2 ijerph-19-10092-f002:**
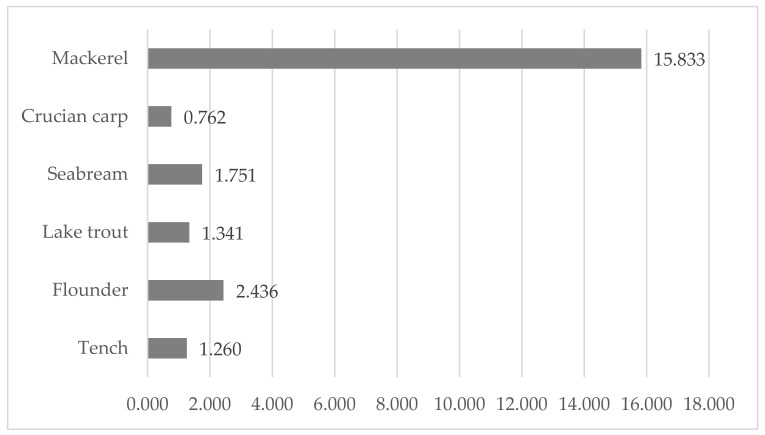
The content of sum of EPA and DHA (mg/g).

**Figure 3 ijerph-19-10092-f003:**
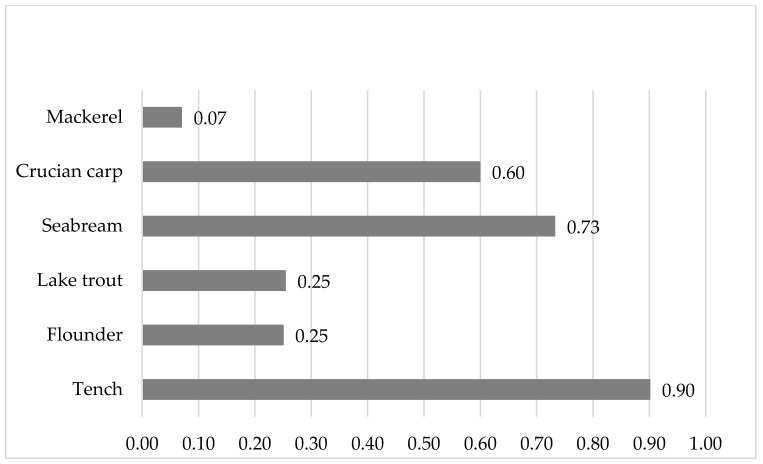
HQ_EFA_ is the hazard quotient for fish consumption when a person obtains from the fish the recommended dose of EFA.

**Figure 4 ijerph-19-10092-f004:**
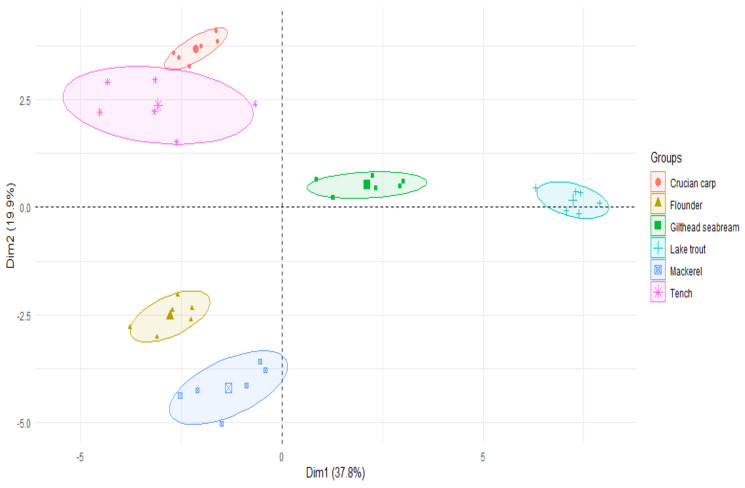
PCA plot, 2D, showing clusterings of fatty acid concentration in six examined fish species.

**Table 1 ijerph-19-10092-t001:** Differences (x ± SD) in the content of mercury (mg/kg wet weight) in muscle tissue of fish and indices (THQ and HI).

	Hg	THQ	HI
Tench (*Tinca tinca* L.)	0.082 ± 0.068 ab	0.1578	
Lake trout (*Salvelinus namaycush* Walb.)	0.024 ± 0.002 c	0.0473	
Flounder (*Platichthys flesus* L.)	0.044 ± 0.021 abc	0.0847	0.684
Gilthead seabream (*Sparus aurata* L.)	0.092 ± 0.008 a	0.1782	
Crucian carp (*Carassius carassius* L.)	0.033 ± 0.007 bc	0.0635	
Mackerel (*Scomber scombrus* L.)	0.079 ± 0.016 ab	0.1525	
RfD (mg/kg/day)		3.00 × 10^−4^	

X—mean; SD—standard deviation; a, b, c —significant differences (*p* ≤ 0.05). The same letter indicates the absence of significant differences h (*p* > 0.05). RfD—oral reference dose (mg/kg/day) [34]; THQ—target hazard quotient; TTHQ—individual foodstuff; HI—hazardous index.

**Table 2 ijerph-19-10092-t002:** Lipid content (%) and fatty acid composition (% of total fatty acids) in muscles of different fish species.

Fatty Acids	Tench		Lake Trout		Flounder		Gilthead Seabream		Crucian Carp		Mackerel	
	mean	SD	mean	SD	$\bar{x}$	SD	$\bar{x}$	SD	$\bar{x}$	SD	$\bar{x}$	SD
*n*	6		6		6		6		6		6	
fat	0.85	0.38	2.15	0.68	1.21	0.54	2.29	0.40	0.76	0.39	8.24	2.31
C12:0	0.17 ab	0.14	0.03 c	0.00	0.06 bc	0.01	0.12 ab	0.04	0.17 a	0.04	0.05 bc	0.03
C14:0	1.45 b	0.49	1.59 b	0.24	2.52 ab	0.28	2.33 ab	0.21	3.80 a	0.50	2.25 ab	0.52
C15:0	0.64 a	0.23	0.15 c	0.01	0.67 ab	0.11	0.28 bc	0.03	1.24 a	0.19	0.46 abc	0.08
C16:0	22.82 a	0.87	8.88 c	0.37	20.26 abc	1.77	17.62 bc	1.23	19.36 abc	2.03	21.31 ab	1.29
C17:0	0.78 ab	0.28	0.14 b	0.02	0.52 abc	0.10	0.28 bc	0.02	1.19 a	0.12	0.45 abc	0.07
C18:0	6.11 a	1.35	2.39 b	0.23	4.25 ab	0.78	4.40 ab	0.50	4.59 ab	0.57	6.12 a	0.46
C20:0	0.20 b	0.02	0.26 a	0.02	0.20 b	0.03	0.25 ab	0.02	0.18 b	0.03	0.35 a	0.09
C22:0	0.00 b	0.00	0.34 a	0.19	0.33 a	0.08	0.17 ab	0.02	0.14 ab	0.19	0.21 a	0.06
Σ SFA	32.17 a	1.73	13.78 c	0.53	28.81 ab	2.38	25.46 b	1.72	30.67 a	3.25	31.20 a	1.66
C14:1	0.07 ab	0.05	0.03 b	0.00	0.07 ab	0.03	0.08 a	0.01	0.23 a	0.03	0.02 b	0.00
C16:1	6.81 b	3.58	3.03 b	0.43	11.85 a	5.81	4.34 b	0.58	11.29 a	1.03	4.38 b	0.55
C17:1	0.84 ab	0.33	0.23 c	0.04	0.68 ab	0.11	0.37 bc	0.03	1.77 a	0.37	0.43 bc	0.05
C18:1cis9	13.93 b	5.31	43.23 a	1.43	14.38 b	3.22	29.02 ab	13.18	19.40 b	1.67	26.64 ab	4.38
C18:1cis9	4.21 ab	0.20	3.00 a	0.27	4.82 ab	0.99	7.56 a	11.66	6.48 b	0.58	4.67 ab	0.23
C20:1 (n-7)	0.16 b	0.04	0.18 b	0.05	2.24 a	0.76	0.09 b	0.01	0.23 ab	0.04	0.50 ab	0.10
C20:1 (n-9)	0.44 bc	0.24	3.84 a	0.50	1.24 bc	0.38	1.59 abc	0.14	1.12 bc	0.28	2.41 ab	0.40
C20:1 (n-11)	0.16 ab	0.16	0.00 b	0.00	1.34 a	0.31	0.00 b	0.00	0.00 b	0.00	0.00 b	0.00
C22:1 (n-9)	0.00 b	0.00	0.47 a	0.07	0.17 ab	0.07	0.30 ab	0.04	0.00 b	0.00	0.68 a	0.10
C22:1 (n-11)	0.00 c	0.00	1.58 a	0.18	0.10 bc	0.02	0.56 ab	0.06	0.07 bc	0.10	0.60 ab	0.32
Σ MUFA	26.62 c	9.12	55.58 a	1.32	36.89 b	5.96	43.90 b	3.86	40.59 b	2.93	40.32 b	4.45
C18:2 (n-6)	6.74 ab	0.56	13.31 ab	0.53	2.39 bc	0.61	14.25 a	0.53	3.39 bc	0.41	0.96 c	0.10
C18:3γ-lin (n-6)	0.35 ab	0.02	0.99 a	1.62	0.25 ab	0.03	0.19 b	0.01	0.47 a	0.02	0.19 b	0.04
C20:2 (n-6)	0.57 abc	0.18	0.99 a	0.21	0.48 bc	0.08	0.59 ab	0.05	0.51 abc	0.09	0.24 c	0.02
C20:3 (n-6)	0.77 a	0.16	0.54 a	0.15	0.07 b	0.02	0.21 ab	0.05	0.31 ab	0.06	0.05 b	0.01
C20:4 (n-6)	7.62 a	2.78	0.46 c	0.13	3.44 ab	0.68	0.84 bc	0.29	2.38 ab	0.52	1.13 bc	0.22
C22:5 (n-6)	1.08 a	0.33	0.31 b	0.13	0.62 ab	0.12	0.51 ab	0.18	0.65 ab	0.11	0.59 ab	0.17
C18:3 (n-3)	3.35 ab	0.80	4.41 a	0.55	0.89 b	0.16	2.56 ab	0.37	5.85 a	0.76	0.65 b	0.22
C18:4 (n-3)	0.24 c	0.10	0.75 abc	0.11	0.87 ab	0.27	0.40 bc	0.09	1.18 a	0.19	1.02 ab	0.37
C20:3 (n-3)	0.38 ab	0.17	0.32 ab	0.05	0.15 bc	0.03	0.20 abc	0.03	0.69 a	0.13	0.11 c	0.02
C20:4 (n-3)	0.50 abc	0.04	0.56 ab	0.05	0.27 c	0.06	0.41 bc	0.08	0.92 a	0.22	0.53 ab	0.07
C20:5 (n-3) EPA	6.67 b	1.83	1.90 c	0.28	12.12 a	2.54	2.85 c	0.80	5.27 b	1.26	6.73 b	1.29
C22:5 (n-3)	2.88 a	0.75	0.91 b	0.17	2.10 a	0.45	1.57 ab	0.47	1.76 ab	0.56	1.70 ab	0.16
C22:6 (n-3) DHA	10.07 b	2.57	5.18 c	0.78	10.64 b	2.43	6.05 c	2.33	5.36 c	1.70	14.58 a	1.45
Σ n-6 PUFA	17.12 a	3.26	16.60 a	0.99	7.24 b	1.17	16.59 a	0.62	7.71 b	0.62	3.17 c	0.42
Σ n-3 PUFA	24.09 a	5.22	14.03 b	1.18	27.06 a	3.43	14.05 b	3.77	21.03 a	4.53	25.31 a	2.72
EPA + DHA	16.75 ab	4.35	7.07 b	0.97	22.76 a	3.41	8.91 b	3.12	10.63 ab	2.89	21.30 a	2.13
n-3/n-6	1.42 c	0.24	0.85 c	0.09	3.78 b	0.50	0.84 c	0.21	2.71 b	0.48	8.10 a	1.30
OFA	24.44 a	0.88	10.50 c	0.46	22.84 ab	1.74	20.07 bc	1.35	23.33 ab	2.48	23.61 ab	1.59
DFA	73.94 b	0.84	88.61 a	0.47	75.44 b	1.97	78.94 ab	1.40	73.93 b	2.97	74.92 b	1.78
AI	0.42 abc	0.03	0.18 c	0.01	0.43 ab	0.04	0.36 bc	0.03	0.50 a	0.08	0.44 a	0.05
TI	0.32 a	0.04	0.16 b	0.01	0.25 ab	0.02	0.34 a	0.08	0.32 a	0.09	0.28 ab	0.01
FLQ	16.75 ab	4.35	7.07 b	0.97	22.76 a	3.41	8.91 b	3.12	10.63 ab	2.89	21.30 a	2.13

SD—standard deviation; a, b, c—significant differences (*p* ≤ 0.05). The same letter (in rows) indicates the absence of significant differences h (*p* > 0.05). EPA—eicosapentaenoic acid (C20:5), DHA—docosahexaenoic acid (C22:6), Ʃ n-6 PUFA—polyunsaturated fatty acid, Ʃ n-3 PUFA—polyunsaturated fatty acid, AI—index of atherogenicity, TI—index of thrombogenicity, FLQ—flesh-lipid quality, DFA—hypocholesterolemic fatty acids, OFA—hypercholesterolemic fatty acids.

## Data Availability

Not applicable.

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
