# Peer review of "Total Mercury and Fatty Acids in Selected Fish Species on the Polish Market: A Risk to Human Health"

_ijerph, 2022, doi:10.3390/ijerph191610092_

Round 1

Reviewer 1 Report

Dear Authors, I have reviewed the manuscript "The effect of total mercury and fatty acids on the quality of marine and freshwater fish on the Polish market. Risk to human health", which aimed to determine the differences between the mercury content in the muscles of various fish species, both marine and freshwater, and to estimate, on the basis of THQ and HI indicators, whether these fish are safe from the nutritional point of view, and thus can be consumed by humans. In addition the aim was to estimate of the fat content, profile of fatty acids and the most important lipid quality indexes in muscles of fish species from Polish markets. Prior to further processing, the following observations and comments should be addressed:

- The first paragraph should be rewritten, the statements are very basic, write in a more technical way about fish and their importance.

- In line 80 it states that fish can accumulate heavy metals, but in spite of this they serve as a source of food... The wording is not correct, if an individual is contaminated by heavy metals for example (methylmercury) you cannot say 'in spite of this, it is a source of food'. Human health should come first and not be played with. 

- Two objectives are described in the last paragraph of the introduction. The way of writing should be improved, you should synthesize by naming the word objective only once (do not repeat the word 'objective').

- In materials and methods you should write an introductory paragraph describing the design, approach and tools used in your research process.

- Include photographs of the species collected at the sales fairs.

- Review the results, which are sometimes discussed in this section.

- In the results be more precise, consider your objectives and try to create a section for each of them. 

- The first paragraph of the discussion is very long. It is necessary to separate it by paragraphs, I recommend you to be more precise and dedicate a paragraph to each established objective. 

- In conclusions, consider the objectives in the same way and dedicate a paragraph to each one of them. Currently the conclusion is very brief

- In the conclusions include the limitations and future research based on your results.

Author Response

Responses to reviews

- The first paragraph should be rewritten, the statements are very basic, write in a more technical way about fish and their importance.

The first paragraph was omitted from the publication. The publication started from the second paragraph.

- In line 80 it states that fish can accumulate heavy metals, but in spite of this they serve as a source of food... The wording is not correct, if an individual is contaminated by heavy metals for example (methylmercury) you cannot say 'in spite of this, it is a source of food'. Human health should come first and not be played with. 

The sentence in line 80 has been deleted from the text.

- Two objectives are described in the last paragraph of the introduction. The way of writing should be improved, you should synthesize by naming the word objective only once (do not repeat the word 'objective').

As suggested by the Reviewer, the word goal was omitted and this part was rewritten.

- In materials and methods you should write an introductory paragraph describing the design, approach and tools used in your research process.

The paragraph suggested by the Reviewer was written in the "Materials and methodology" section.

- Include photographs of the species collected at the sales fairs.

At the time of purchase of the fish, no photos were taken, as it is forbidden (not allowed) on the sales market.

- Review the results, which are sometimes discussed in this section.

As suggested by the Reviewer, in the "Results" section, sentences that should be included in the "Discussion" were crossed out and moved to this section.

- In the results be more precise, consider your objectives and try to create a section for each of them. 

As suggested by the Reviewer, the "Results" section was divided into parts consistent with the research objectives.

- The first paragraph of the discussion is very long. It is necessary to separate it by paragraphs, I recommend you to be more precise and dedicate a paragraph to each established objective. 

As suggested by the Reviewer, the "Discussion" section was divided into paragraphs.

- In conclusions, consider the objectives in the same way and dedicate a paragraph to each one of them. Currently the conclusion is very brief

As suggested by the Reviewer in the "Conclusion" section we tried to take into account all the goals and devote a separate paragraph to them. We hope that the conclusions were properly described.

- In the conclusions include the limitations and future research based on your results.

As suggested by the Reviewer, comments were taken into account in the "Conclusions" section.

Reviewer 2 Report

The manuscript is interesting but there are some issues to be considered. The authors needs to revise the manuscript before its publication.

1) Title: "The effect of total mercury and fatty acids on the quality of marine and freshwater fish on the Polish market. Risk to human health."

The manuscript is about the effect of mercury and fatty acids on the quality? Or is about the toxic risks? Please, revise this title because is not adequate.

2) Introduction: Are you explained about the effects of arsenic, however, As has not determined in the manuscript. Please, revise the introduction and just include the issue of this research.

3) Samples: include the samples and the characteristics in a table. Origin, package, ...

4) The number of samples is not adequate. Just 36 samples?

5) Results and discussion: should be revised. Is difficult to understand and to follow correctly.

6) Conclusions: include a guideline recommendations to the population

Author Response

Responses to reviews

1) Title: "The effect of total mercury and fatty acids on the quality of marine and freshwater fish on the Polish market. Risk to human health."

The manuscript is about the effect of mercury and fatty acids on the quality? Or is about the toxic risks? Please, revise this title because is not adequate.

As suggested by the Reviewer, the title has been changed.

2) Introduction: Are you explained about the effects of arsenic, however, As has not determined in the manuscript. Please, revise the introduction and just include the issue of this research.

As suggested by the Reviewer, sentences concerning arsenic have been removed from the "Introduction" section.

3) Samples: include the samples and the characteristics in a table. Origin, package, ...

The fish that were purchased on the market were fresh whole or gutted (see methodology) and had no packaging.

The origin of the fish is also presented in the text.

4) The number of samples is not adequate. Just 36 samples?

According to statisticians' guidelines, six samples are sufficient for statistical analysis.

5) Results and discussion: should be revised. Is difficult to understand and to follow correctly.

We hope that, as suggested by the Reviewer, the results and discussion have been revised.

6) Conclusions: include a guideline recommendations to the population

As suggested by the Reviewer, the conclusions have been changed.

Round 2

Reviewer 1 Report

I have reviewed the new version and the authors have complied with the requested changes.